# Increased T-bet/GATA-3 and ROR-γt /Foxp3 Ratios in Cerebrospinal Fluid as Potential Criteria for Definite Neuro-Behçet’s Disease

**DOI:** 10.3390/jcm11154415

**Published:** 2022-07-29

**Authors:** Meriam Belghith, Olfa Maghrebi, Aroua Cherif, Khadija Bahrini, Zakaria Saied, Samir Belal, Samia Ben Sassi, Mohamed-Ridha Barbouche, Mariem Kchaou

**Affiliations:** 1Laboratory of Transmission, Control and Immunobiology of Infections, Institut Pasteur de Tunis, Tunis 1002, Tunisia or olfamaghrebi3@gmail.com (O.M.); bahrini_khadija@live.fr (K.B.); ridha.barbouche@pasteur.rns.tn (M.-R.B.); 2Department of Biology, Tunis El Manar University, Tunis 1068, Tunisia; 3Faculty of Medicine, Tunis El Manar University, Tunis 1007, Tunisia; 4Neurological Department, Charles Nicolle Hospital, Tunis 1006, Tunisia; arouachrif@yahoo.fr (A.C.); docmariem@yahoo.fr (M.K.); 5Neurology Department, Mongi Ben Hamida National Institute of Neurology, Tunis 1007, Tunisia; zakariasaied@hotmail.com (Z.S.); samir_belal@ymail.com (S.B.); bensassisam@yahoo.fr (S.B.S.)

**Keywords:** neuro-Behçet’s disease, T-bet/GATA-3, ROR-γt/Foxp3, cerebrospinal fluid, PBMCs

## Abstract

When the central nervous system (CNS) is the primary affected site in an initial attack of Behçet’s disease (BD), the differential diagnosis is particularly challenging. Some cases remain unclassified or qualified as probable neuro-Behçet’s disease (NBD). Several cytokines are involved in the immunopathogenesis of this disease; however, studies establishing the differential cytokine pattern between probable and definite NBD are scarce. Twenty-eight parenchymal NBD patients, diagnosed according to the International Consensus Recommendation (ICR) criteria and classified into definite (D-NBD; *n* = 17) and probable (P-NBD; *n* = 11), were sampled at their first neurological symptoms, and compared with healthy control subjects (*n* = 20). Oligoclonal bands (OCB) of IgG were detected by isoelectric focusing on agarose, and immunoblotting of matched serum and cerebrospinal fluid (CSF) sample pairs. T cell cytokines (INF-γ, IL-4, IL-17, and IL-10) and transcription factors related to Th1, Th2, Th17, and T regulatory populations (respectively T-bet, GATA-3, ROR-γt, and Foxp3) were studied by quantitative RT-PCR in peripheral blood mononuclear cells (PBMCs) and CSF cells. Inflammatory cytokines such as IL-6, TNF-α, and IL-1β were also analyzed. CSF OCB pattern 2 was present in only 1 out of 28 neuro-Behçet’s patients who belonged to the P-NBD group. Two D-NBD patients had OCB in CSF showing pattern 4. In the D-NBD CSF samples, IL-17 and IL-10 expressions were significantly elevated compared to P-NBD. Moreover, D-NBD patients had increased levels of T-bet/GATA-3 and ROR-γt/Foxp3 ratios compared to P-NBD. Furthermore, a significant increase of CSF IL-6 in D-NBD, compared to P-NBD and the controls, was found. In addition to the increased IL-6 level, the data obtained suggest the existence in D-NBD patients of a significantly disrupted balance between Th17 effector and T regulatory cells, as reflected by the enhanced ROR-γt/Foxp3 ratio. This could be considered as an additional criterion for definite neuro-Behçet’s disease.

## 1. Introduction

Behçet’s disease (BD) is a chronic relapsing, multisystem vasculitis. Diagnosis of BD is based on the International BD Study Group (ISG) Criteria [1]. It is established on the basis of the presence of recurrent oral aphthous ulcerations and two of the following criteria: genital ulcerations, skin lesions, eye lesions, such as cells in the vitreous and retinal vasculitis, and a positive pathergy test.

One of the specific forms of this disease is neuro-Behçet’s disease (NBD), which is characterized by neurological involvement, and is a cause of long-term morbidity and mortality [2]. Although BD is a well-defined disease, with well-established criteria for its diagnosis [1], this is more challenging for NBD. Among NBD patients, parenchymal and non-parenchymal central nervous system (CNS) involvement defines two separate entities [3,4]. Parenchymal NBD is the most commonly seen form, comprising 60 to 75% of cases, and affecting the brainstem, mesodiencephalic junction, cerebellar peduncles, and cerebral hemispheres. Lesions are typically characterized by areas of T2 prolongation on brain magnetic resonance imaging (MRI), which have been suggested to represent a small vessel vasculitis [5].

The prevalence of CNS involvement varies widely, from 1.3% to 59%, depending on diagnosis criteria and ethnic background [2,6,7]. Epidemiological studies show that BD is the most frequent vasculitis in Tunisia [8], and the frequency of NBD is 28.1% among BD patients [9]; it appears within 5 years of the onset of BD symptoms. These particular forms can mimic other CNS inflammatory diseases, especially multiple sclerosis, which is also prevalent in our country [10]. In 2014, the clinical, laboratory, and neuroimaging features of “definite” and “probable” NBD symptoms were established by a panel of neuro-Behçet syndrome (NBS) experts [11]. Due to the absence of specific biomarkers, the diagnosis of definite NBD requires a high index of clinical suspicion, laboratory, and neuroimaging findings in an individual who fulfills the diagnosis criteria for BD [11]. However, diagnosis remains probable according to the International Consensus Recommendation (ICR) in two cases: suggestive neurological syndrome as in definite NBD, with systemic BD features but not satisfying International Study Group (ISG) criteria, or a non-characteristic neurological syndrome occurring in the context of ISG criteria-supported BD [11]. In these cases, treatment is challenging, due to the occurrence of neuronal loss at early stages of these potentially disabling diseases.

NBD etiology remains unclear, but it implicates genetic and environmental factors, largely involving immunological effects. The immunological response is associated with a non-specific inflammatory reaction which implicates pro- and anti-inflammatory cytokines secretion [1,12,13,14]. Among the cytokines implicated in the pathogenicity of the disease [15], IL-6 has been reported to be markedly elevated in the CSF of NBD patients, and to correlate with the disease activity [16]. Furthermore, the balances between Th1/Th2 and Th17/Treg have been shown to play a role in different autoimmune and inflammatory diseases, including Behçet’s disease [17]. However, data concerning peripheral blood and cerebrospinal fluid (CSF) levels of cytokines in NBD patients are limited [1,12,13,14]. Borhani and colleagues showed different patterns between parenchymal and non-parenchymal subdivisions of NBD [1]. Cytokine differences were also described between acute and chronic progressive parenchymal NBD [13,18]. To our knowledge, whether cytokine levels significantly differ among definite versus probable NBD patients has never been studied. Accordingly, here we carried out a clinical and laboratory study, in order to describe the profile of cytokines and T-cell subsets transcription factors in peripheral blood mononuclear cells (PBMCs) and CSF of NBD patients classified into definite and probable, aiming to identify a marker which could confirm definite NBD diagnosis compared to a group of probable NBD patients and the controls.

## 2. Materials and Methods

### 2.1. Study Population

An exploratory cross-sectional study was conducted on 28 patients with a diagnosis of NBD, according to the International Consensus Recommendation (ICR) criteria [11], recruited at the Neurological Department of Charles Nicolle Hospital and Neurology Service of Mongi Ben Hamida National Institute (Tunis, Tunisia). Figure 1 shows a flow chart for the selection of the study population. 

Inclusion criteria were as follows: age ≥ 18 years old; diagnosis of parenchymal forms of NBD classified into definite (*n* = 17) and probable (*n* = 11). Patients with vascular-NBD, as well as subjects with comorbid/coexisting disorders, were excluded from this study. Patients with relapses, who had been treated with high doses of steroids less than 1 month before enrollment, were excluded. No patient had received any immunomodulatory or immunosuppressive treatment for at least 3 months before the study and prior to the sample collection. No patient had a history of recent vaccination before inclusion in the study. Other CNS infections and inflammatory disorders were excluded.

Definite NBD (D-NBD) were patients with BD (ISG+) under colchicine treatment who developed neurological signs (*n* = 10), or patients with first neurological symptoms suggestive of NBD, who developed later systemic signs fulfilling ISG criteria (ISG+) (*n* = 7). Probable NBD (P-NBD) were patients with first neurological symptoms suggestive of NBD but remaining ISG− after 3–5 years of follow-up (*n* = 7), or patients (ISG+) who developed neurological symptoms but non-characteristic neurological syndrome of NBD (*n* = 4). All these patients were diagnosed and sampled at the active stage of the disease (at the onset of neurological features). Then, the patients were treated with high doses of steroids and pulses of cyclophosphamide added 6 months later by Azathioprine.

A full review of each patient’s clinical history was conducted. Demographic, epidemiological, and clinical data, as well as neuroimagery, laboratory tests, and therapeutic management, were recorded. Other possible etiologies had been excluded by immunologic testing, serology, and neuroimaging. The control group consisted of 20 individuals with persistent headaches requiring lumbar puncture, to exclude meningitis or meningeal hemorrhage, with normal magnetic resonance imagery (MRI). 

The project was approved by the Ethical Committee of the Pasteur Institute of Tunis, and written informed consent was obtained from all participants before their inclusion in the study.

### 2.2. Blood and CSF Samples

CSF samples were collected from 48 subjects (17 D-NBD, 11 P-NBD, and 20 controls). Peripheral blood mononuclear cells (PBMCs) were analyzed for 45 subjects (15 D-NBD, 10 P-NBD, and 20 controls). PBMCs were isolated using the Ficoll technique, then conserved in TRIzol Reagent (Sigma-Aldrich, Taufkirchen, Germany), and stored at −80 °C for RNA extraction.

CSF cells were obtained after centrifugation at 1400 rpm for 10 min; the cells platelets were then resuspended in lysis buffer RLT from RNeasy Mini Kit (Qiagen, Venlo, The Netherlands), supplemented by beta-mercaptoethanol, and conserved at −80 °C.

### 2.3. CSF Albumin and Immunoglobulin Analysis

Albumin and IgG concentration were measured in the CSF and sera. IgG index albumin ratios were calculated using the following formulae for the IgG index (IgG CSF/IgG serum)/(Albumin CSF/Albumin serum). Isoelectric focusing on agarose and immunoblotting of matched serum and CSF sample pairs identified 4 characteristic oligoclonal bands (OCB) patterns [19,20]. Type1 was a normal pattern where no bands were identified. Type 2 indicated intrathecal synthesis, where bands were seen only in the CSF. When the pattern of bands seen was identical in both sera and CSF, a “mirror pattern” type 4 pattern was recorded. Identical shared bands, but additional CSF specific bands, indicated a type 3 pattern.

### 2.4. RNA Isolation and cDNA Synthesis

Total RNA was extracted from the PBMCs and CSF cells using the RNeasy Mini Kit (Qiagen), according to the manufacturer’s protocol. The samples were treated with DNAse, in order to avoid DNA contamination. RNA purity and concentration were determined using NanoDrop. The first-strand cDNA was synthesized for each RNA sample using a High-Capacity cDNA Reverse Transcription Kit (Applied Biosystems^TM^ by Thermo Fisher, Vilnius, Lithuania). This reaction was performed with the following parameters: 1 µg RNA for 25 °C for 10 min, 37 °C for 2 h, and 85 °C for 5 min.

### 2.5. Quantitative Real-Time PCR

Quantitative real-time PCR was carried out using SYBR Green technology on the Applied Biosystems ABI PRIZM 7500 Real-Time PCR System. All samples were run in duplicate, and relative quantification of mRNA levels was performed using glyceraldehyde-3-phosphate dehydrogenase (GAPDH) as an endogenous reference. The reaction volume of 20 μL was amplified for 40 cycles with the following parameters: 95 °C for 15 s and 60 °C for 1 min. The CSF and PBMCs expressions of cytokines (IFN-γ, IL-1β, TNF-α, IL-17, IL-6, IL-4, and IL-10) and transcription factors (T-bet, GATA-3, RoR-γt, and Foxp3) were assessed. Primers, as previously reported [21,22], were used for real-time PCR. HPLC-purified oligonucleotides primers were bought from CarthaGenomics Advanced Technologies (Tunis, Tunisia).

### 2.6. Statistical Analysis

Quantitative data were expressed as the mean ± SD. Statistical significance was determined by Graphpad Prism version 8 (GraphPad, San Diego, CA, USA), using a non-parametric Mann–Whitney test to compare between two groups. Results were considered statistically significant where the *p* value < 0.05. 

## 3. Results

### 3.1. Demographic Features

Data relative to the demographic and clinical characteristics of the subjects included in the study are summarized in Table 1. There were no significant differences in age distribution among the two groups of patients (D-NBD and P-NBD) and the controls. Indeed, age at onset of neurological signs was 35.76 for D-NBD and 37.45 for P-NBD. The mean interval between neurological and systemic manifestations was 5.57 months for D-NBD patients. For the group of P-NBD with neurological symptoms at first onset (*n* = 7), none of them had developed systemic BD signs after a period of follow-up of 50 months. Neurological signs were developed after a mean interval of 31.2 months in patients classified D-NBD, and of 13.5 months in P-NBD. The mean period of follow-up of all NBD patients was 49.5 months.

### 3.2. Neurological Characteristics, MRI Findings, and CSF Analysis

The neurological characteristics, MRI findings, and CSF analysis of the patients are reported in Table 2. Among a variety of neurological manifestations, motor and pyramidal signs were the most frequent signs in the two groups of NBD patients. Isolated supratentorial location was present in 46.4% of all NBD patients. In addition, brainstem lesions were observed more in D-NBD patients (53% of D-NBD versus 36% in P-NBD), and myelitis was observed only in one case of P-NBD. CSF analysis showed a significant increase (*p* = 0.0004) in the median of the CSF protein levels in the two groups of patients (D-NBD = 0.60 g/L; P-NBD = 0.63 g/L), as compared to the controls (0.39 g/L). We also noted a significantly elevated CSF cell count in the group of D-NBD as compared to P-NBD (D-NBD = 32.6/mm^3^; P-NBD = 23/mm^3^; *p* = 0.017). These two groups of patients showed a significantly marked elevation of cell count as compared to the controls (controls = 4.3/mm^3^; *p* < 0.0001). None of the patients had an increased IgG index, while OCB were present only in one patient with P-NBD disease (profile 2), and in two D-NBD patients (profile 4).

### 3.3. Cytokines Expression 

In order to evaluate the cytokine profile in the two groups of D-NBD and P-NBD, we first compared the expression of different cytokines and transcription factors related to Th1, Th2, Th17, and Treg in the PBMCs and CSF of D-NBD (Group 1), P-NBD (Group 2) patients, and a group of controls. In the PBMCs samples (Figure 2), we found no significant difference in the mean of the studied transcription factors (T-bet, GATA-3, RoR-γt, and Foxp3) and T cells subsets cytokines (IFN-γ, IL-4, IL-17, and IL-10) between the two groups of NBD patients. Interestingly, in this compartment (PBMCs), we noticed a significantly increased level of INF-γ mRNA in PBMCs of P-NBD, as compared to the controls (*p* < 0.0001) (Figure 2a).

In the CSF compartment (Figure 3), comparisons among the three different groups indicated that IFN-γ expression showed a significant increase in NBD patients compared to the controls (*p* = 0.022; *p* < 0.0001). For IL-17, we noticed a significant increase in patients compared to the controls (*p* < 0.0001 D-NBD vs. controls; *p* = 0.036; P-NBD vs controls). The expression of this cytokine was also significantly increased among the D-NBD group as compared to P-NBD (*p* = 0.0093). We also analyzed the ROR-γt expression, the master transcription factor of Th17 cells, in the CSF of the three studied groups. As shown in Figure 3c, the CSF of the D-NBD patients had higher expression levels of ROR-γt than the controls (*p* = 0.039).

IL-4 and GATA-3, the two factors thought to govern the Th2 response, were also investigated, but no significant differences were detected between the three studied groups (Figure 3b). We also investigated the differential expression of IL-10 anti-inflammatory cytokine between patients and the controls. Our results showed that this cytokine was significantly increased in the two groups of NBD compared to the healthy control group (*p* = 0.28; *p* = 0.0002). Interestingly, a significantly enhanced expression of IL-10 in D-NBD, as compared to P-NBD, was observed (*p* = 0.045) (Figure 3d). Moreover, the study of the relative expression of the Foxp3 transcription factor in the CSF showed a significantly higher expression of Foxp3 in the P-NBD (median = 2.06) group, as compared to D-NBD (median = 1.097) (*p* = 0.0262).

Intending to uncover the balance between effector and regulatory T cells in the CSF compartment, we proposed to study the transcription factors ratio of Th1/Th2 and Th17/Treg cells (Figure 4). This analysis revealed an increase in T-bet/GATA-3 (Th1 vs. Th2) (*p* = 0.0011) and ROR-γt/Foxp3 (Th17 vs. Treg) ratios in D-NBD patients, as compared to P-NBD (*p* = 0.0149). 

To better determine the pro-inflammatory response in the PBMCs and CSF of the patients, we evaluated mRNA expression of IL-6, IL-1β, and TNF-α. We found no significant difference in the expression of TNF-α and IL-1β between the three studied groups (Figure 5). However, concerning IL-6 expression, we found a significant increase in all NBD patients compared to the controls (P-NBD vs. controls *p* = 0.0003 and D-NBD vs. controls *p* < 0.0001). Furthermore, CSF IL-6 expression was higher in D-NBD patients compared to P-NBD (*p* = 0.0184) (Figure 5a).

## 4. Discussion

The main aim of the present study was to evaluate a differential cytokine profile in the PBMCs and CSF of NBD classified as definite versus probable parenchymal NBD. The most striking findings from this study were the increased T-bet/GATA-3 and ROR-γt/Foxp3 CSF ratios in D-NBD patients. Moreover, CSF IL-6 mRNA expression could be considered as a discriminative marker between probable and definite NBD. Although the key role of IL-6 has already been pointed out in neuro-Behçet’s disease by several investigations [23,24], in this current study we show that IL-6 expression in CSF can support definite NBD diagnosis, as compared to probable NBD. 

The demographic and clinical characteristics illustrated in this finding are in line with previous studies [25,26,27]. Indeed, it has been reported that the prevalence of NBD disease was two times more frequent in men than in women [2,4]. The mean duration of BD before neurological manifestation onset was ≅2.5 years (26.14 months) for patients included in this study, and ranged from 3 to 6 years in previously described studies [3,28,29,30]; however, the neurological presentation could occur simultaneously with the first systemic symptoms of BD or precede them (6% of patients) [28,29]. Sometimes, diagnosis of P-NBD could be advanced in high prevalence areas, and treatment with immunosuppressive drugs could be started. In 14 cases of inaugural neurological signs, 50% of patients (*n* = 7) developed systemic symptoms such as buccal and genital aphthosis, pseudofolliculitis, and erythema nodosum. These symptoms fulfilled the ISG criteria, and the patients became D-NBD after a mean period of 5.57 months. The remaining 50% of patients (*n* = 7) did not develop systemic signs, suggesting BD during 3 to 5 years of follow-up. All the patients included in this study received the same treatment, except the four BD patients with non-characteristic neurological signs. 

By neuroimaging, it has been shown that relevant syndromes in NBD patients include brainstem syndrome, multiple-sclerosis-like presentations, movement disorders, meningoencephalitic syndrome, myelopathic syndrome, cerebral venous sinus thrombosis (CVST), and intracranial hypertension [2,5,31,32]. In the present study, 8/17 (47.06%) of D-NBD patients had isolated supratentorial location, and 9/17 (52.94%) D-NBD patients had brainstem lesions.

The CSF of patients included in our study showed pleocytosis, which was in accordance with previous reports on CSF cellularity in parenchymal NBD patients [3,33]. Few studies have investigated the CSF immunoelectrophoretic data in NBD. In a report performed on 121 NBD patients, it was demonstrated that only 8 patients had OCB in CSF showing pattern 2 [34]. All these positive cases had parenchymal NBD. These observations argue in favor of the scarce presence of intrathecal oligoclonal IgG bands in the CSF of NBD patients. In our study, OCB in CSF showing pattern 2 was found in a single patient. This patient was classified as P-NBD, and did not develop BD signs or ISG criteria. On the other hand, 25/28 (89.29%) subjects had pattern 1, and two D-NBD patients had pattern 4, suggesting that oligoclonal IgG were synthesized in the blood, and migrated across impaired blood-CSF barriers showing mirror pattern bands. The most important differential diagnosis, in this case, was a chronic infectious disease like tuberculosis or brucellosis, which occur especially in endemic areas. The two patients with profile 4 had non-infectious meningoencephalitis-like presentation. 

It has been shown that cytokines and inflammatory mediators play an important role in the pathogenesis of BD. In the CSF, our data indicated a significant increase of IFN-γ and IL-17 in both P-NBD and D-NBD, as compared to the controls. Indeed, in Behçet’s disease, an exacerbated inflammation due to IFN-γ and IL-17 secretion has been described in skin and brain lesions [35]. Moreover, it has been reported that these two cytokines are involved in the persistence and the progression of the disease [36]. However, other reports did not show any difference in the secretion of IFN-γ between NB groups and patients with headaches attributed to Behçet’s disease and the controls [1]. More interestingly, a significantly increased level of INF-γ mRNA observed in the PBMCs of P-NBD, as compared to the controls, was consistent with previous reports, which described an increased IFN-γ level in BD patients with active uveitis [37]. 

A previous study reported by Hamzaoui, et al., has described an enhanced expression in NBD patients of RoR-γt (Th17) and T-bet (Th1), and a low expression of GATA-3 (Th2) and Foxp3 [38]. The increased ROR-γt/Foxp3 and T-bet/GATA-3 ratios in the CSF of NBD, compared to headache attributed to BD (HaBD) and NIND patients, suggests the activation of inflammatory T cell populations and the dysregulation of T helper cells in the CSF of NBD patients. In the current study, we report an increase in T-bet/GATA-3 and ROR-γt/Foxp3 ratios in the CSF of definite, as compared to probable, NBD, which could be considered as novel criteria discriminating these two forms of the disease.

Among the cytokines involved in the pathogenicity of the disease, CSF IL-6 has been emphasized as a marker of disease activity and long-term outcome in heterogeneous groups of NBD patients [2,8]. It has also been underlined that IL-6 is more prominently increased in chronic progressive NBD patients. Moreover, Borhani, et al., have detected an increased level of IL-6 in patients with parenchymal forms, as compared to HaBD and healthy controls [1]. Furthermore, it has been demonstrated that IL-6 and IL-8, but not IL-1β or TNF-α, are elevated in CSF obtained from acute or chronic progressive NBD by ELISA [13]. In agreement with these observations, we found no significant difference in the expression of IL-1β and TNF-α between definite and probable NBD patients in the two studied compartments. 

Furthermore, another member of the IL-1 family, IL-33, was described as upregulated in NBD compared to HaBD and NIND patients, and this higher expression was correlated with IP-10 and MCP-1 chemokines [14]. In previous studies, Hamzaoui, et al., showed that elevated CSF levels of IL-15 in patients with NBD, in comparison with BD in remission and HC, were reported. Similarly, serum IL-15 levels were found to be increased in active NBD. IL-15 was involved in the BD inflammatory process, particularly in vasculitis foci, as an elevated CSF/serum IL-15 ratio characterizes vascular cerebral lesions [14,39]. 

The same authors showed significant elevation of CSF IL-6 in patients with progressive NBD compared to patients with active BD without neurological manifestations [24]. Moreover, it was noted that CSF, but not serum samples of NBD patients with acute parenchymal NBD, displayed significantly increased IL-6 levels, as compared to other groups [40]. Wang, et al., showed that the IL-6 level, which was higher in CSF of NBD patients, dropped when the disease activity subsided [16]. IL-6 was also used as a marker of therapeutic response in a Japanese center [41], and was shown to be significantly lower in patients with more favorable outcomes [42].

Additionally, we showed a higher level of IL-10 mRNA expression in D-NBD patients, as compared to P-NBD CSF, despite the decreased foxp3 expression in this group. We speculate that this IL-10 was not associated with regulatory cells. In line with this observation, a recent report of our team has suggested that this cytokine is elevated in naive treatment NBD patients [21,43]. These data are in accordance with results from Aridogan, et al., who found an increased level of IL-10 in sera of active Behçet’s disease [15]. 

## 5. Conclusions

Collectively, these results showed elevated ratios of T-bet/GATA-3 and ROR-γt/Foxp3 in the CSF of definite NBD patients, indicating a disturbed balance toward the predominance of T effector cells and an inflammatory process. Furthermore, our data indicate that IL-6 could be a potential discriminative biomarker for definite NBD. We also showed the absence of intrathecal OCB synthesis in D-NBD patients during the first episode of neurological relapse. A better understanding of the pathogenesis of borderline forms is needed among immunologic comparative studies between forms of NBD and other CNS inflammatory diseases. Our finding is an attempt to unravel differential NBD immunopathogenesis that could explain heterogeneous clinical onset. Further studies, with a wider range of patient samples, are needed to determine the impact of treatment on disease progression and the secretion of these mediators. 

## Figures and Tables

**Figure 1 jcm-11-04415-f001:**
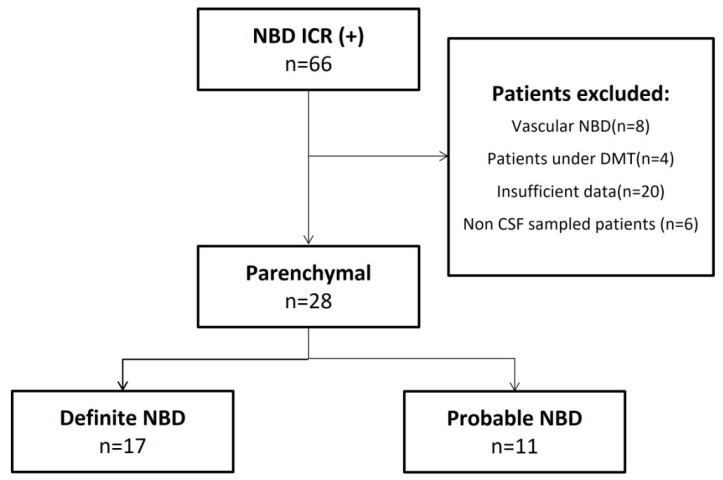
Flow chart depicting patient selection and grouping. DMT: disease-modifying therapy; CSF: cerebrospinal fluid; ISG: International Study Group.

**Figure 2 jcm-11-04415-f002:**
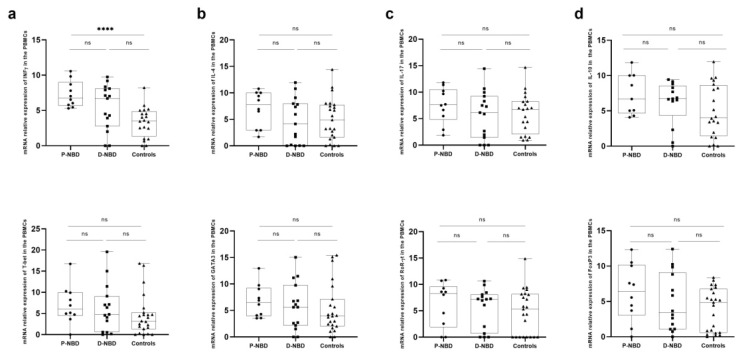
Boxplots representation of PBMCs Th1, Th2, Th17 and Treg related gene expression: (**a**) Representative boxplots of INF-ɣ and T-bet expression in the PBMCs of patients and NIND; (**b**) Representative boxplots of IL-4 and GATA3 expression in the PBMCs of patients and NIND; (**c**) Representative boxplots of IL-17 and RoR-ɣt expression in the PBMCs of patients and NIND; (**d**) Representative boxplots of IL-10 and Foxp3 expression in the PBMCs of patients and NIND. Statistical significance between the two groups was assessed using the Wilcoxon–Mann–Whitney test ns = non significant, **** *p* < 0.0001.

**Figure 3 jcm-11-04415-f003:**
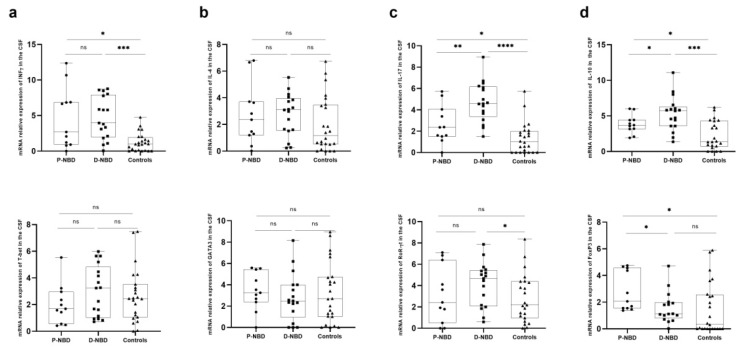
Boxplots representation of CSF cells’ Th1, Th2, Th17 and Treg-related gene expression: (**a**) Representative boxplots of INF-ɣ and T-bet expression in the CSF of patients and NIND; (**b**) Representative boxplots of IL-4 and GATA3 expression in the CSF of patients and NIND; (**c**) Representative boxplots of IL-17 and RoR-ɣt expression in the CSF of patients and NIND; (**d**) Representative boxplots of IL-10 and Foxp3 expression in the CSF of patients and NIND. Statistical significance between the two groups was assessed using the Wilcoxon–Mann–Whitney test ns = non significant, * *p* ≤ 0.05, ** *p* < 0.01, *** *p* < 0.001,**** *p* < 0.0001.

**Figure 4 jcm-11-04415-f004:**
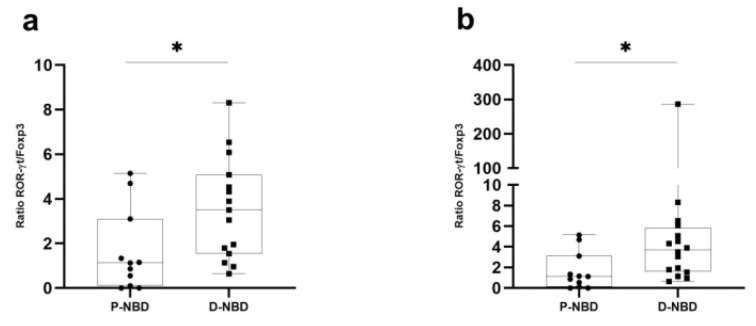
Boxplots representation of CSF ratios of the selected transcription factors: (**a**) Representative boxplots of RoR-ɣt/ Foxp3 ratio in the CSF of P-NBD and D-NBD; (**b**) Representative boxplots T-bet/GATA3 ratio in the CSF of P-NBD and D-NBD. Statistical significance between the two groups was assessed using the Wilcoxon–Mann–Whitney test * *p* ≤ 0.05.

**Figure 5 jcm-11-04415-f005:**
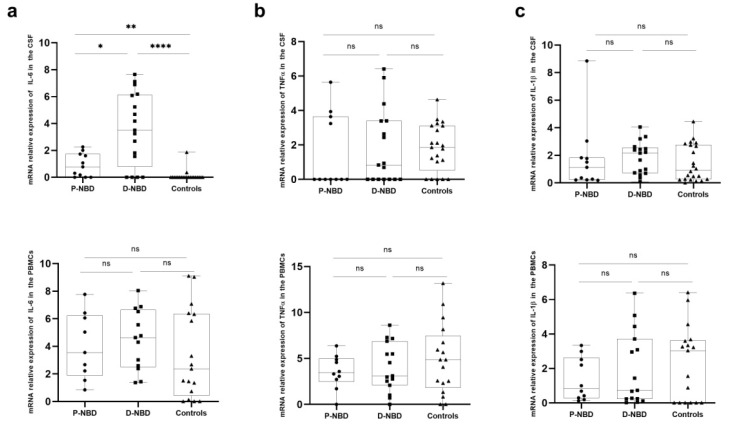
Boxplots represention of inflammation-related genes expression in PBMCs and CSF cells: (**a**) Representative boxplots of IL-6 expression in the PBMCs and CSF of patients and NIND; (**b**) Representative boxplots of TNFα expression in the PBMCs and CSF of patients and NIND; (**c**) Representative boxplots of IL-1β expression in the PBMCs and CSF of patients and NIND. Statistical significance between the two groups was assessed using the Wilcoxon–Mann–Whitney test ns = non significant, * *p* ≤ 0.05, ** *p* < 0.01, **** *p* < 0.0001.

**Table 1 jcm-11-04415-t001:** Demographic and clinical characteristics of subjects included in the study.

	Overall (Patients)	Definite NBD	Probable NBD	Controls	*p* Value
*n* = 48	*n* = 28	*n* = 17	*n* = 11	*n* = 20	
Gender (F/M)	12/16	8/9	4/7	16/4	0.0312
Age (mean ± SD)	40.66 (±10.39)	41 (±11.14)	40.2 (±9.80)	42.52 (±20.96)	0.8941
Patients with BD first (ISG+)	*n* = 14	*n* = 10	*n* = 4		0.0754
Interval between BD and onset of neurological signs (months)	26.14 (±22.02)	31.2 (±16.21)	13.5 (±12.54)		
Patients with neurological signs first	*n* = 14	*n* = 7	*n* = 7		
Interval between neurological signs and BD (months)	-	5.57 (±3.95)	-		
Age at onset of neurological signs (mean ± SD)	35.25 (9.74)	35.76 (9.62)	37.45 (9.00)		0.5716
Follow-up period (months) (mean ± SD)	49.5 (±13.4)	50.46 (±14.2)	49 (±13.2)		0.7870

**Table 2 jcm-11-04415-t002:** Neurological characteristics, MRI findings, and CSF data of subjects included in the study.

	Overall Patients (*n* = 28)	Definite NBD (*n* = 17)	Probable NBD (*n* = 11)	Controls (*n* = 20)	*p* Value
Neurological characteristics:				-	
- Headaches	8	4	4		
- Hemiparesis/pyramidal signs	22	14	8		
- Sensory symptoms	12	6	6		
- Cranial nerve involvement	9	5	4		
- Ataxia	6	5	1		
- Psychiatric signs	5	1	4		
- Seizures	1	1	0		
MRI findings				-	
- Isolated supratentorial location	13	8	5		
- Basal ganglia	9	5	4		
- Internal capsule	7	2	5		
- Brainstem lesions	14	9	4		
- Myelitis	1	0	1		
- No abnormalities	2	0	2		
CSF analysis					
- CSF protein concentration (g/L)	…	0.60 (+0.21)	0.63 (+0.19)	0.39 (+0.13)	0.0004
- Cell count (/mm3)	…	32.6 (+7.32)	23.7 (+7.66)	4.3 (+5.6)	<0.0001
- IgG Index	…	0.59	0.495	0.49	0.4979
- IEPP					
Type 1	25	15	10	20	
Type 2 and 3	1	0	1	0	
Type 4	2	2	0	0	

## Data Availability

The data presented in this study are available on request from the corresponding author.

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
