# Peer review of "Increased T-bet/GATA-3 and ROR-γt /Foxp3 Ratios in Cerebrospinal Fluid as Potential Criteria for Definite Neuro-Behçet’s Disease"

_jcm, 2022, doi:10.3390/jcm11154415_

Round 1

Reviewer 1 Report

In this manuscript, the authors investigated potential biomarkers for the purpose of differential diagnosis between definite NBD and probable NBD.

-        Overall, the manuscript is well written with clear language.

-        Minor comments:

o   The brief conclusion on page 1 contains new results. The conclusion should only include the take-home message from the manuscript.

o   On page 10 line 311 the authors mentioned patient 8 showed pattern 2. That was the first time this expression “pattern 2” was introduced. This needs more clarification and elaboration from the authors to define this term.

o   It would more interesting not to phrase the conclusion in points on page 12 line 376 – 379.

o   Of note as well that what is written in the conclusion on page 12 line 376 – 379 is a repetition of what is written on the same page line 317 – 374. 

Author Response

We wish to express our appreciation to the reviewers for their invaluable comments and suggestions that really improved the quality of our manuscript.

Our point-by-point answers are given below. Our answers are mentioned right after each comment. Accordingly, the changes we have made in the text are indicated in red. We believe we have addressed all of the minor comments that were raised by the reviewers.

We really hope that this revised version will be found acceptable for publication in Journal of clinical medicine. We are looking forward to hear from you.

Reviewer reports:

Reviewer #1:

-        Minor comments:

o   The brief conclusion on page 1 contains new results. The conclusion should only include the take-home message from the manuscript.

Thank you for this comment, we agree that this sentence could be misleading so we introduced some clarification in the conclusion as follow: Besides the increased IL-6 level, the data obtained suggest the existence in D-NBD patients of a significantly disrupted balance between Th17 effector and T regulatory cells reflected  by the enhanced  ROR-γt/Foxp3 ratio.

o   On page 10 line 311 the authors mentioned patient 8 showed pattern 2. That was the first time this expression “pattern 2” was introduced. This needs more clarification and elaboration from the authors to define this term.

Thank you for this recommendation. In the material section, we have introduced the  4 characteristic oligoclonal bands (OCB) patterns which are identified by Isoelectric focusing on agarose and immunoblotting of matched serum and CSF samples. Type1 is a normal pattern where no bands are identified. Type 2 indicates intrathecal synthesis, where bands are seen only in the CSF.  When the pattern of bands seen is identical in both sera and CSF, a « mirrored » type 4 is recorded. Identical shared bands but additional CSF specific bands indicate a type 3 pattern. To be more precise, we added two references explaining the 4 existing patterns. Page 4, Line 141.

 Andersson M, Alvarez-Cermeño J, Bernardi G, Cogato I, Fredman P, Frederiksen J, et al. Cerebrospinal fluid in the diagnosis of multiple sclerosis: a consensus report. J Neurol Neurosurg Psychiatry. 1994. August; 57(8): 897–902.

 Freedman MS, Thompson EJ, Deisenhammer F, Giovannoni G, Grimsley G, Keir G et al. Recommended standard of cerebrospinal fluid analysis in the diagnosis of multiple sclerosis: a consensus statement. Arch Neurol. 2005. June; 62(6): 865–70 doi: 10.1001/archneur.62.6.865

o   It would more interesting not to phrase the conclusion in points on page 12 line 376 – 379.

Thank you for this suggestion, we improved the conclusion as follows: Collectively, these results showed elevated ratios of T-bet/GATA-3 and ROR-γt/Foxp3 in the CSF of definite NBD patients indicating a disturbed balance toward the predominance of T effector cells and an inflammatory process. Furthermore, our data indicate that IL-6 could be a potential discriminative biomarker for definite NBD. We also showed the absence of intrathecal OCB synthesis in D-NBD patients during the first episode of neurological relapse. Page 12, Line 370-375.

o   Of note as well that what is written in the conclusion on page 12 line 376 – 379 is a repetition of what is written on the same page line 370 – 374.

We completely agree with this comment, we do delete the redundancy on line 370 – 374. We hope that these modifications improved the manuscript and render it more suitable

Reviewer 2 Report

Herein, the assessment of cytokine profile in specific cohorts of blood (PBMCs) and CSF of in Neuro-Behçet patients was performed in a small patient’s cohort. The authors using biochemical and molecular approaches concluded in raised rations of T-bet/GATA-3 and ROR-γt/Foxp3 in CSF in definite NBD patients as well as increased of specific transcripts between patients and controls. In general, the provided information is satisfactory. From a scientific point of view, I have a few comments and suggestions:

I suggest the abstract could be better organized as one single text highlighting the importance of new findings.

Line 130: PBMCs samples were collected from 45 or 25 (15 D-NBD and 10 P-NBD) subjects? Please explain.

Line 264: The authors proposed IL-6 as candidate marker for definite NBD. However, IL-6 expression does not shown in Figure 5 (Fig 5c is not included). The authors should correct the Figure.

How immunosuppressive treatment could affect the profile of T-bet/GATA-3 ratio and interleukins levels?

Regarding immunoblotting analysis, the authors could provide a characteristic picture of agarose gels and specific bands.

Author Response

We wish to express our appreciation to the reviewers for their invaluable comments and suggestions that really improved the quality of our manuscript.

Our point-by-point answers are given below. Our answers are mentioned right after each comment. Accordingly, the changes we have made in the text are indicated in red. We believe we have addressed all of the minor comments that were raised by the reviewers.

We really hope that this revised version will be found acceptable for publication in Journal of clinical medicine. We are looking forward to hear from you.

Reviewer reports:

Reviewer #2: 

I suggest the abstract could be better organized as one single text highlighting the importance of new findings.

Thank you for this suggestion. We organize the abstract as one single text.

Line 130: PBMCs samples were collected from 45 or 25 (15 D-NBD and 10 P-NBD) subjects? Please explain.

Thank you for this comment. As suggested this sentence was corrected as follows:  for 45 subjects (15 D-NBD, 10 P-NBD and 20 controls). Page 4, Line 130-131.

Line 264: The authors proposed IL-6 as candidate marker for definite NBD. However, IL-6 expression does not shown in Figure 5 (Fig 5c is not included). The authors should correct the Figure.

We do apologize for this mistake. We corrected the figure accordingly.

How immunosuppressive treatment could affect the profile of T-bet/GATA-3 ratio and interleukins levels?

We do agree with this important comment as immunosuppressive treatment will affect the profile of cytokines and  T-bet/GATA-3. However, the aim of this article is the comparison between definite and probable neuro-behçet who haven't received any immunomodulatory or immunosuppressive treatment for at least 3 months before the study, prior to the sample collection. However, we are actually conducting a comparative analysis between treated and untreated neuro-behçet patients and we have only preliminary results showing a tendency of IL-6 decrease.

Regarding immunoblotting analysis, the authors could provide a characteristic picture of agarose gels and specific bands.

Thank you for this suggestion. Isoelectric focusing gel are performed for all the studied patients and is a routine analysis that we choose not to include in the manuscript. However, we show you below the result of the patient included in the article with pattern 2 as compared to one patient with pattern 1.
